# Spontaneous symmetry breaking and panic escape

**Choong Sun Kim**[1⊛]**, Claudio Dib**[2⊛]**, Sechul Oh**[3⊛*]

**1** Department of Physics and IPAP, Yonsei University, Seoul, Korea, **2** Department of Physics and CCTVal, Federico Santa Maria Technical University, Valparaíso, Chile, **3** University College, Yonsei University, Incheon, Korea

⊛ These authors contributed equally to this work.
* scohph@yonsei.ac.kr

## Abstract

Panic-induced herding in individuals often leads to social disasters, resulting in people being trapped and trampled in crowd stampedes triggered by panic. We introduce a novel approach that offers fresh insights into studying the phenomenon of asymmetrical panic-induced escape. Our approach is based on the concept of Spontaneous Symmetry Breaking (SSB), a fundamental governing mechanism in the Physical Sciences. By applying the principles of SSB, we conjecture that the onset of disastrous effects of panic can be understood as a SSB phenomenon, and we formulate the process accordingly. We highlight that this way of understanding panic escape leads to simple general measures of preventing catastrophic situations, by considering two crucial parameters: *population density* and *external information*. The interplay of these two parameters is responsible for either breaking or restoring the symmetry of a system. We describe how these parameters are set by design conditions as well as crowd control. Based on these parameters, we discuss strategies for preventing potential social disasters caused by asymmetrical panic escape.

## 1 Introduction

Panic [1] is an intense surge of fear that overwhelms the ability to think clearly and logically, overtaking it with acute anxiety, uncertainty, and frenzied agitation akin to a fight-or-flight response. It can strike individuals unexpectedly or emerge abruptly among large groups as mass panic, which is closely linked to herd behavior. Panic, as a specific type of collective behavior, arises in contexts where resources are scarce or diminishing, such as in situations with *high group density* and *limited communication*. Individuals in a state of panic exhibit maladaptive and persistent collective actions, including blocking and dangerously overcrowding, behaviors thought to spread through social contagion.

Helbing et al. [2] provide an overview of the typical characteristics associated with escape panics. These features include individuals moving or attempting to move much more rapidly than usual, the formation of congestion or blockages among people, a tendency toward collective behavior, and a failure to consider alternative exits in emergency situations. The occurrences observed in panic situations often differ significantly from those in normal

**Data availability statement:** All relevant data are within the manuscript.

**Funding:** This work was supported by grants from the National Research Foundation (NRF) of the Korean government (RS-2022-NR074767),

(RS-2022-NR070836). We acknowledge support from Chile grants Fondecyt 1210131 and ANID PIA/APOYO AFB230003. The funders had no role in the study design, data collection and analysis, decision to publish, or preparation of the manuscript.

**Competing interests:** The authors have declared that no competing interests exist.

circumstances. Panic frequently triggers crowd stampedes, leading to serious social catastrophes [3–5]. One recent tragic instance of a panic-induced disaster is the Halloween crush that occurred among panicked individuals in South Korea [5].

The panic phenomenon has been the subject of numerous systematic studies, both in theory and experiment. Notable theoretical studies include the works of Helbing et al. [2], Burstedde et al. [6], Tajima and Nagatani [7], Kirchner and Schadschneider [8], and Perez et al. [9]. On the experimental front, researchers such as Saloma et al. [10], Helbing et al. [11], and Altshuler et al. [12]. have contributed to this area of research.

In particular, Helbing et al. [2] introduced a theoretical framework for the dynamics of panic founded on a generalized (physical-social) force model that considers the collective behavior of panic escape as self-propelled many-particle systems. They explored scenarios involving congestion arisen from panic by simulating the dynamics of pedestrian crowds, predicting the phenomenon of panic-induced symmetry breaking in situations where pedestrians attempt to escape from a smoke-filled room with two exits. In a detailed formulation the authors proposed a dynamical approach to explain the panic escape behavior, involving various interaction forces in equations of motions as [2]:

$$m_i \frac{d\vec{v}_i}{dt} = m_i \frac{v_i^0(t)\vec{e}_i^{\,0}(t) - \vec{v}_i(t)}{\tau_i} + \sum_{j(\neq i)} \vec{f}_{ij} + \sum_W \vec{f}_{iW}. \tag{1}$$

Here $m_i$ and $\vec{v}_i$ are the mass and velocity of the pedestrian $i$, respectively. Also, $v_i^0$ and $\vec{e}_i^{\,0}$ denote a certain desired speed and direction, while $\tau_i$ is a certain characteristic time of the pedestrian $i$, respectively. The interaction forces $\vec{f}_{ij}$ and $\vec{f}_{iW}$ contain many free parameters as well:

$$\vec{f}_{ij} = \left\{ A_i \exp[(r_{ij} - d_{ij})/B_i] + kg(r_{ij} - d_{ij}) \right\} \vec{n}_{ij} + \kappa g(r_{ij} - d_{ij}) \Delta v_{ji}^t \vec{t}_{ij}, \tag{2}$$

$$\vec{f}_{iW} = \left\{ A_i \exp[(r_i - d_{iW})/B_i] + kg(r_i - d_{iW}) \right\} \vec{n}_{iW} - \kappa g(r_i - d_{iW})(\vec{v}_i \cdot \vec{t}_{iW}) \vec{t}_{iW}. \tag{3}$$

For the detailed definition of all these parameters, we refer to Ref. [2]. As seen in the above equations, this detailed dynamical approach involves so many free parameters that predictability becomes quite small without some simplifying assumptions. Therefore the reduction of the number of independent parameters is desirable in order to increase predictability. (Predictability can be defined as "the number of observables minus the number of parameters".) In the study conducted by Altshuler et al. [12], experiments were conducted using escaping ants as a model for pedestrians to demonstrate and validate the predictions made by Helbing et al. [2].

In this work we conjecture that the phenomenon of panic-induced escape can be considered and formulated as a case of spontaneous breakdown of a symmetry of the system. Spontaneous Symmetry Breaking (SSB) is a well known mechanism that occurs in several other phenomena of collective behavior in the physical sciences. Moreover, due to its general character, the formulation of SSB, instead of trying to formulate in detail the dynamics of each individual component of the system, exploits the *symmetries* of the system in terms of some global or average variables which exhibit such symmetries. The formulation of a multiparticle system in terms of a global variable that represents the average of the system is sometimes called a *mean field approximation*. In terms of such a global variable, the asymmetric behavior can be explained with the concept of *spontaneous symmetry breaking* (SSB) [13,14], or *hidden symmetry*, where the laws that govern the system have some symmetry, but this symmetry is not manifested in the actual states of the system.

We will show that panic escape phenomena, which are known to exhibit an asymmetric behavior, can be formulated in terms of the SSB mechanism, where the onset of the asymmetric behavior arises from two critical factors in a crowd of individuals: *high population density* and *limited external information*. We will show how specific combinations of these two factors lead to either the breakdown or restoration of symmetry in such a system.

The structure of this paper is as follows: In Sect 2 we quote an experiment done with a population of ants that is representative of panic escape of crowds, and show that it exhibits the mechanism of spontaneous symmetry breakdown. In Sect 3 we give a brief overview of the SSB concept and its general formulation. In Sects 4 and 5, we apply the SSB formulation to the case of panic in crowds of ants that was quoted in Sect 2. In Sect 6 we elaborate on the variables that could be controlled to avoid catastrophic situations caused by panic responses of crowds, and comment on strategies to prevent potential social disasters arising from asymmetrical panic-induced escape. Finally in Sect 7 we state our conclusions.

## 2 Panic-induced escape: The ants experiment

The phenomenon of herding is a common feature of collective behavior observed in various species during panic situations, including humans. Theoretical predictions [2] have suggested that when individuals are in a confined space during a panic situation, it can lead to an asymmetrical use of two identical exit doors. A study conducted by Altshuler et al. [12] provides experimental evidence of the occurrence of asymmetrical panic escape. In this study, ants were used as a model to represent pedestrians, as shown in Fig 1. In high panic conditions, the majority of ants preferred one exit over the other, even when the latter was more available, resulting in a "symmetry-broken" or asymmetrical behavior. Conversely, in low panic conditions, the ants exhibited nearly equal usage of both exits, approaching a "symmetric" pattern. This observation demonstrates the panic-induced asymmetry in ant escape behavior.

The phenomenon of panic tends to occur in crowded areas or large groups during emergencies where, due to anxiety and confusion, individuals exhibit a common behavior by imitation. Panic situations may exhibit the following characteristics: (1) Group behavior: In panic situations, individuals tend to follow similar behavioral patterns within a group; this is

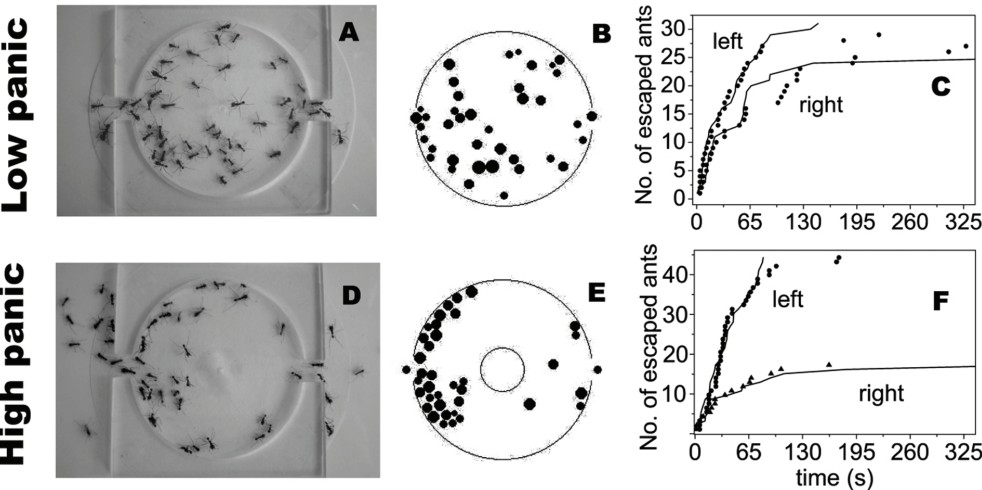

**Fig 1. Ants escaping from a chamber with two exits symmetrically positioned under low and high panic conditions, adapted from Ref [12].**

commonly observed in large events or emergency situations. (2) Neglect of alternative exits: Despite the presence of alternative exits, individuals in panic situations may ignore them and move all in a specific direction. (3) Increased speed: In panic situations, individuals attempt to move faster than usual, which can increase the likelihood of accidents.

## Possible sources of panic

Panic can occur in various situations and the causes are diverse; moreover these causes may interact with each other, and measures must be taken in advance for prevention, or for mitigation and management once panic sets on. The sources of panic can be broadly categorized into two groups: external (extra) panic sources and internal (intra) panic sources.

External panic sources include: sudden emergency situations such as fires or smoke hazards [15,16], seismic events like earthquakes [17], water-related risks [18], incidents by chemicals like repellents, pheromones [12,19], airborne toxins or gases [20,21], incidents by physical elements like temperature [22,23] or lighting conditions [24], risks by biological factors such as ants [25] or wasps, and violent incidents; large-scale events such as crowded events or festivals with physical barriers or obstacles [26–32], psychological triggers such as fear, anxiety or impatience [33]; loud noises and chaos; the exchange of information or signals among individuals through various forms of signage or communication [34–42].

In contrast, internal or inner sources causing panic include: high population density [2,15, 18,24,26,27,33,35–37,43–46]; large total size of a group or degree of herding [47–50]; mass movements such as delays in public transportation or traffic congestion; group behavior in situations of heightened disorder or chaos.

## Common issues in previous research

As mentioned above, numerous studies have been conducted on panic phenomena caused by various panic sources. However, previous studies have approached and analyzed specific panic phenomena using methods tailored to those phenomena, applying specific theories and methods. Because panic sources are numerous and diverse, and the behaviors resulting from different sources can vary, each had to be approached separately. For instance, studies on panic phenomena caused by fire have been conducted using approaches tailored to those specific phenomena, while panic phenomena such as bank-run have been analyzed using approaches tailored to those specific ones. Considering the numerous cases for different sources such as human behavior, fire, bank runs, and so on, and approaching each one separately, leads to an endless and unmanageable situation. An approach that can comprehensively cover all panic phenomena in a general way and provide an integrated and complete picture of the overall panic phenomenon is desirable both in theoretical and practical terms.

## Fundamental solution to these issues

As observed in panic situations of crowds of people in a stadium or ants in a confined place, a common feature that consistently appears when the disastrous escape sets in, is the breaking of a symmetry in the escaping flow. For example, in the ant experiment mentioned earlier, when panic occurs in the two-exit experiment, the symmetry is broken as more ants tend to crowd towards one of the two exits. This could be the result of evolutionary or unknown psychological reasons, but the consistent feature in all panic phenomena is the breaking of symmetry in this manner. After noticing this symmetry-breaking feature, one can ask why this symmetry breakdown occurs, and moreover, what variables determine

the onset of this symmetry breakdown in the panic situations. We conjecture that this spontaneous symmetry-breaking behavior in panic escape is a general feature of panic scenarios, and it can be formulated in a similar way as it is done for many physical systems. For example, it is well known that spontaneous symmetry breaking (SSB) occurs in cases such as a pen standing vertically on its tip on a flat horizontal surface, or in ferromagnetic materials below the Curie temperature. Therefore, the breaking of a symmetry observed in panic phenomena can be easily connected to the concept of SSB that we describe in the next section.

We will show that a valuable understanding of panic-induced escape phenomena as the one described above can be achieved by considering two fundamental factors which, from a dynamics point of view, can be considered as "forces": *population density* related to internal sources and denoted by a parameter $\rho$, is a "force" that tends to make the individuals act according to the behavior of their neighbors, and *external information* connected to external sources and represented by a parameter $f$, is a "force" that tends to make the individuals respond according to external conditions or instructions. In those terms, a larger $f$ tends to keep the symmetry in case of panic, while a larger $\rho$ tends to break the symmetry and lead to uncontrolled or even disastrous escape under panic. For instance, the phenomenon of ants escaping under high panic conditions occurs when there is a combination of *high* density $\rho$ and very *little* external information $f$ available among them.

In the following section, we will explore how these two factors, $\rho$ and $f$, are related to the mechanism of SSB.

## 3 The concept of spontaneous symmetry breaking

In this section we provide a concise overview of the core concept of *spontaneous symmetry breaking* (SSB) [13,14], which will be a key idea of the present work to explain the panic-escape phenomenon.

In Physics, a *symmetry* is the name of a transformation that leaves invariant the governing rules (laws of motion) of a physical system. For example, if we consider the motion of a ball inside a hemispherical bowl, the governing laws of the system are invariant under rotations around a vertical axis (see Fig 2).

The symmetry does not mean that the motion itself (a state of the system) has to be invariant under the transformation, but it means that it will transform into another, similar motion with the same energy; for example, if the ball oscillates back and forth in a given vertical plane, a rotation of the system around the vertical axis implies that the ball would oscillate just the same but in a different vertical plane. However the *ground state* (the state of minimal energy or the most stable state), which in this case corresponds to the ball at rest at the bottom of the bowl, *is* invariant under rotations. In such case, where not only the laws are invariant but the ground state is invariant as well, we say that the symmetry is explicit.

In contrast, *spontaneous symmetry breaking* (SSB) refers to the case where a system's ground state does *not* exhibit the symmetry of the laws of motion: if we apply the transformation, we obtain another ground state, different than the previous one and not symmetric either, but otherwise equivalent as the previous one. Since what one directly observes in the physical world is the state, not the laws that govern it, we say that the symmetry is *hidden* or *spontaneously broken*, due to the arbitrary selection of a ground state that shows no symmetry.

An example similar to the above of the ball inside the bowl, but which exhibits SSB, is that of a pen standing vertically on its tip on a flat horizontal surface (see Fig 3). The laws of this system are also invariant under a rotation around the vertical axis, however ground state is

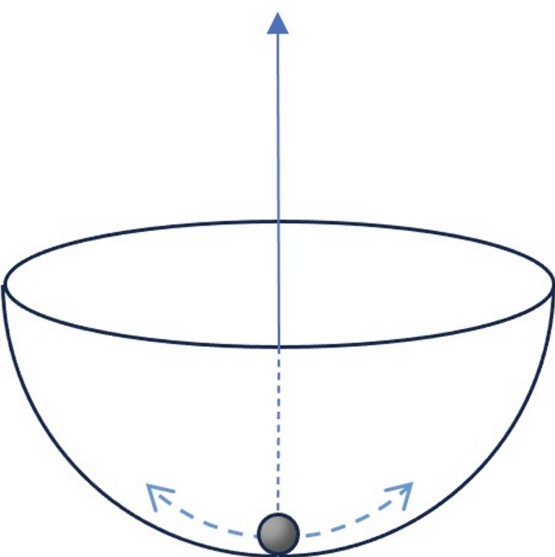

**Fig 2. An example of a system with symmetry.** A ball is moving inside a hemispherical bowl. The system is invariant under rotations around the vertical axis. The oscillating motion of the ball on any given vertical plane would transform to a similar oscillating motion but on another vertical plane. However, the state of minimal energy (ball at rest at the bottom of the bowl) is unique and invariant under the symmetry transformation.

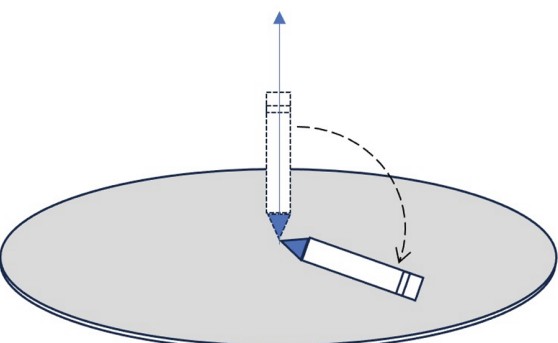

**Fig 3. An example of SSB.** A pen standing vertically on its tip is unstable: to reach its minimal energy it will fall into a horizontal position, thus pointing into some direction. Although the system has a symmetry around the vertical axis (any direction is equivalent), the state of minimum energy breaks that symmetry.

not. Indeed, a state which is invariant under rotations is the pen standing in a vertical position; however this state is not of minimal energy but unstable: in its ground state, it will lie horizontally on the surface. It can lie in any direction, all of them equivalent, but whatever that direction is, it breaks the symmetry: the state of minimal energy does not exhibit invariance under rotations around a vertical axis. Such rotations will transform one ground state into a different one, albeit with the same energy. Many other well-known examples of SSB exist in Nature, such as a ferromagnetic material below the Curie temperature [51] where a spontaneous direction of magnetization appears.

In this study, we demonstrate that panic behaviors, such as ant escape during panic, bank runs, and panic-induced stampedes, naturally result from SSB. Before we go into the specifics

of SBB for panic escape, let us describe a formulation common to many SSB phenomena. For this purpose we just need two ingredients: a global variable that reflects the symmetric or asymmetric state of the system (sometimes called the *order parameter*), which here we generically denote as $\phi$, and a potential energy function (or more generally a *cost function*) $V(\phi)$ that the dynamics of the system seeks to minimize.

Analogous to mechanical systems where the forces push towards the minimum of the potential energy, in a broad sense the cost function represents some quantity, property or sensation that the individuals seek to minimize through their actions. The actions of the individuals (moving in crowds, trying to escape, etc) change the value of the order parameter, which will end up settling at the minimum of the cost function, $V(\phi)$. The value of $\phi$, which represents the state of the system, should also reflect the condition of symmetry, either manifest or hidden (spontaneously broken). More specifically, the symmetry of the system must be a transformation of the state $\phi$ that leaves the laws of the dynamics of the system, i.e. $V(\phi)$, invariant. In the case of the ants that escape to the right or to the left (considering only a one-dimensional system), the symmetry transformation can be chosen to be a sign change $\phi \to -\phi$, and $V(\phi)$ must be invariant under it: $V(\phi) = V(-\phi)$. In order to characterize SSB, it is enough to consider a polynomial in $\phi$ with quadratic and quartic terms only:

$$V(\phi) = (\rho_c - \rho)\phi^2 + f\phi^4, \tag{4}$$

where we have called $(\rho_c - \rho)$ and $f$ the respective coefficients. Now, for this formulation to describe our ants system, we need to give meaning to each of these quantities.

As mentioned above, the cost $V(\phi)$ represents an undesirable or unpreferred perception that the ants seek to minimize and, as in most SSB formulations, we conjecture that $V(\phi)$ has a quartic and a quadratic term. The quartic term, $f\phi^4$, has a minimum at $\phi = 0$ and, as $\phi$ moves away from zero, grows symmetrically for $\phi < 0$ and $\phi > 0$. This term represents some external rule, law, or information that the ants would prefer to obey, i.e. minimize its cost, by keeping $\phi$ as close to zero as possible. The strength of this rule or information is parameterized by the constant $f$. On the other hand, the quadratic term $(\rho_c - \rho)\phi^2$, represents another cost, related to a message or interaction the ants receive from their neighbors. The parameter $\rho$ represents the density of ants in the system, or similarly the intensity of the effect of nearest neighbors on a given ant. The coefficient of the quadratic term is written as $(\rho_c - \rho)$, so that the constant $\rho_c$ is the critical value of the density parameter $\rho$, for which the quadratic term changes sign: for densities below $\rho_c$ the quadratic term is minimal at $\phi = 0$, just as the quartic term, but for densities above $\rho_c$ the quadratic term is maximal at $\phi = 0$ and lower costs are reached for values away from the symmetric point $\phi = 0$. This is the case of SSB: the symmetry is spontaneously broken; a high population density in a panic situation causes an asymmetric stampede, where the ants prefer to follow their neighbors instead of the external information.

In this way, if $\rho < \rho_c$, both costs are minimal at $\phi = 0$, and the ants will tend to distribute symmetrically around the center of the region (see Fig 4, Left). Under a need to leave the region, they will migrate symmetrically to the left and right exits. In the sense of costs, the ants will feel more comfortable (less costly) by obeying the global information set by $f$.

However, if the density $\rho$ happens to be larger than the critical value $\rho_c$, something different happens: the quadratic term has lower costs as $\phi$ moves away from zero, and one finds that the minimum cost $V$ is found for two non-zero values $\phi = \pm\phi_0 \neq 0$, where

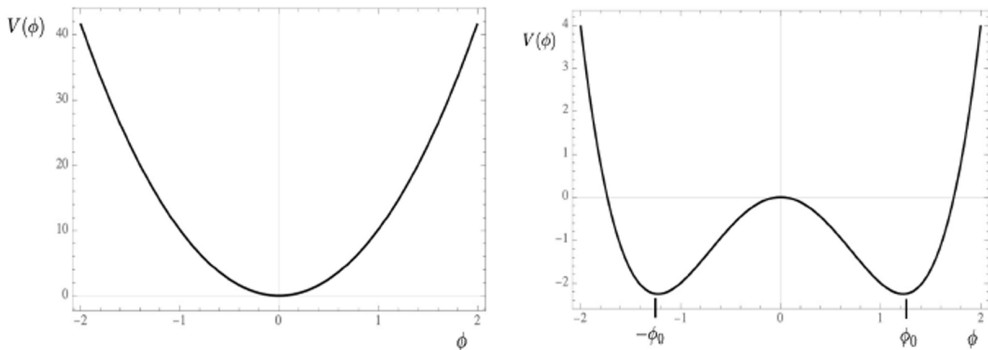

**Fig 4. Cost function $V(\phi)$ in two scenarios.** Left: symmetric scenario where $\rho < \rho_c$ and therefore the minimum of $V$ is at $\phi_0 = 0$. Right: broken symmetry scenario where $\rho > \rho_c$ and therefore $\phi_0 \neq 0$. The scales are in arbitrary units.

$$\phi_0 \equiv \sqrt{\frac{(\rho - \rho_c)}{2f}}, \quad \text{when } \rho > \rho_c, \tag{5}$$

and $\phi_0 = 0$ when $\rho \leq \rho_c$ (or when $\rho/f \to 0$ if $\rho_c = 0$).

In this case, the effect of a large $\rho$ (i.e. many near neighbors) overrides the tendency of global information given by $f$ and, in panic, the ants follow the local crowd creating an asymmetry in their distribution. When $\phi$ settles at the minimum of $V$, either at $+\phi_0$ or $-\phi_0$, it is said that the system shows *spontaneous symmetry breaking*: the physical laws that govern the system are *symmetric* or invariant under a transformation, which in this case is the *reflection* $\phi \to -\phi$, but the state where the system settles does not exhibit the symmetry (see Fig 4, Right). As can be also seen in Fig 4 and Eq (5), by making the value $(\rho - \rho_c)/2f$ smaller we tend to restore the symmetry, i.e. the asymmetrical $V(\phi)$ becomes near symmetric.

Notice that we are considering an example of SSB in 1 dimension (along a line), where the escape options are necessarily discrete (left or right). Nevertheless, the generalization of SSB to 2 dimensions (i.e. any direction of 360° in a circle) is straightforward. Also notice that a SSB scenario could only occur in systems which have a priori a symmetry, that could be broken spontaneously. Therefore, to study SSB we must consider the external factors that cause the panic escape not to be asymmetric, *e.g.*, there is no fire on one specific side, because then obviously the individuals will run in the opposite direction. We are assuming the external factors that cause the panic to be a priori symmetric.

In summary, in a SSB scenario, the physical laws of the system, in the SSB case the physical laws in this system, which are given by the cost function $V(\phi)$, satisfy the symmetry $V(\phi) = V(-\phi)$, but the state where the system settles (i.e. the way the world looks like) is not symmetric: although the ant configurations with $\phi = \phi_0$ or $\phi = -\phi_0$ are equivalent in terms of minimum cost, they are not the same configuration and neither of them exhibits the symmetry: instead, the transformation takes us from one non-symmetric configuration to the other. That is precisely what characterizes a system with spontaneous symmetry breaking (SSB). Up to here our state variable $\phi$ indicates whether we are in a symmetric or spontaneously broken phase, but it is still an abstract variable.

To complete the formulation of SSB applied to the ant stampede, we need to define the physical meaning of the state variable $\phi$, which we do in the following section.

## 4 Panic escape in terms of SSB

As discussed earlier, the natural phenomenon of panic observed in crowds has diverse sources. Therefore, when attempting to apply scientific analysis to the actual phenomenon of panic, a large number of detailed conditions are required. To understand a general picture of the panic phenomenon rather than considering all the detailed conditions that appear in each case, we formulate the scenarios by considering only the most fundamental and core variables, which are: (i) an order parameter $\phi$ that describes the state of the system, and (ii) two parameters that determine the dynamics, namely the crowd density $\rho$, and the external information $f$. In what follows, we illustrate the phenomenon of SSB formulated in terms of the variables $\phi$, $\rho$ and $f$, in a panic situation where the motion of the individuals is along one dimension, and the escape routes are on the left and right ends only, just as in the ant experiment described in Sect 2.

Before we further describe the situation of panic escape in terms of SSB, please note that we only conjecture the onset of disastrous effects of panic through SSB phenomena, and we formulate the process accordingly, instead of describing the onset by a specific dynamics of forces generated among panicked members and surroundings. As is well known in physics, any SSB cannot be induced or described by forces of dynamics but generated spontaneously in nature. Notice that many evacuation studies show symmetry breaking as a phenomenon that emerges from evacuation behavior: whether in panic or not, sometimes the evacuees' behavior causes symmetry breaking, even though they do not intentionally try to cause it – i.e., symmetry breaking occurs spontaneously, even though there is nothing a priori that causes such asymmetry. Therefore SSB can only be caused by a natural tendency of crowds to follow the behavior of neighbors. This tendency will cause asymmetry even when the geometry of the environment is symmetric. Our conjecture is that the appearance of SSB may not necessarily mean a panic or disaster situation, but it is indicative that the crowd is following neighbor's behavior instead of obeying external dynamical rules. In panic situations this behavior could lead to disaster, so it will be valuable to determine general conditions for the onset of SSB in crowds located in otherwise symmetric environments. The power of describing panic as a SSB phenomenon is its simplicity and generality, which does not depend on the various sources of panic nor on the different situations encountered, as explained in Sect 2. As an example, for the reduction of the number of independent parameters, which may be an essential part of solving Eq (1), the understanding of the panic in terms of SSB can help naturally to find the most essential parameters.

We want to formulate the problem in a simple but representative way, by describing the state of the system with a single variable $\phi$, the "order parameter", yet to be identified with some physical quantity of the system formed by a crowd of e.g. ants. Let us consider a one-dimensional scenario where the ants are confined to a segment of the X axis, $x \in [-L, L]$ and assume there are exits at both ends (possibly narrow exits that the ants would jam if they try to exit all at once).

The distribution of ants in the region can be described by a function $n(x)$, so that $n(x)dx$ is the number of ants between $x$ and $x + dx$. Let us assume, for illustration purposes, that the initial distribution is gaussian with average position $\bar{x}$ and width $\sigma$:

$$n(x) = \frac{N}{\sqrt{2\pi\sigma^2}} e^{-\frac{(x-\bar{x})^2}{2\sigma^2}} . \tag{6}$$

Here $N$ is the total number of ants if we assume that the region is large enough ($L \gg \sigma$):

$$N = \int_{-L}^{L} n(x)\, dx, \tag{7}$$

and the average position $\bar{x}$ corresponds to

$$\bar{x} = \frac{1}{N} \int_{-L}^{L} n(x)\, x\, dx. \tag{8}$$

A full dynamic treatment would be necessary to determine the time evolution of the distribution, $n(x,t)$, subject to the interaction of each of the ants with the external information as well as the interactions among one another (neighbor interactions). In such treatment one should consider ants moving with different velocities, which could continually change due to the interactions among them. However, as a simplification we want to treat the panic behavior in terms of a single "mean field", similarly as to how ferromagnetism is treated by means of an average magnetization, even though it is the average result of a miriad of atomic magnets continually interacting with each other and flipping directions. In our case, the mean field that describes collectively the ants behavior will be the net flux $\phi$ (which we just call *flux* for simplicity):

$$\phi = \int n(x)\, v(x)\, dx, \tag{9}$$

where $v(x)$ is the velocity of the ants in the region $x \to x + dx$.

To simplify local details further, let us assume that all the ants move at the same speed $v$, some of them to the right ($v_x = +v$) and the others to the left ($v_x = -v$). (A continuous distribution of speed would characterize better some situations than using a common speed $v$, however, the main feature that the SSB is addressing remains the same: does the crowd follow external pre-established rules or is it dominated by nearest neighbor behavior?) Let us then call $n_R(x)$ and $n_L(x)$ the distributions of *right-moving* ants and *left-moving* ants, respectively, so that the total distribution is

$$n(x) = n_R(x) + n_L(x), \tag{10}$$

while the flux is:

$$\phi = \int \left( n_R(x) - n_L(x) \right) v\, dx \quad = (N_R - N_L)\, v. \tag{11}$$

Here $N_R$ and $N_L$ are the total number of right-movers and left-movers, respectively. We will consider that the ants get the urge to escape at $t = 0$ and start moving at their chosen speed from then on. This consideration corresponds to a model where the ants, in average, do not change their direction once they start moving, and so $N_R$ and $N_L$ are constant from the moment the ants are in the need to escape. In this way, if $n_R^{(0)}(x)$ and $n_L^{(0)}(x)$ are the distributions of right-movers and left-movers at $t = 0$, at later times these distributions will simply be:

$$n_R(x,t) = n_R^{(0)}(x - vt), \qquad n_L(x,t) = n_L^{(0)}(x + vt). \tag{12}$$

From here it is easy to show that the average position of the ants as a function of time for $t > 0$ is

$$\bar{x}(t) = \bar{x}(0) + \frac{\phi}{N} t, \tag{13}$$

at least until the time they reach the exits, when a slowdown or even jamming may occur. Here we are only interested in the onset of the panic behavior and not on the consequences, so we will disregard slowdown or jamming at the exits.

Let us now consider the symmetry of the problem and the spontaneous breaking of it. We start at $t = 0$ with a distribution $n(x)$ which is symmetric, i.e. $n(x) = n(-x)$. Now consider that at that instant the ants have the urge to escape, each one moving with a speed $v$, either to the left or to the right.

A rigorous definition of symmetry is that the whole distribution should remain symmetric for any $t > 0$, i.e. $n(x,t) = n(-x,t)$. This symmetry condition occurs if and only if the distributions of right- and left-movers are functions that satisfy the relation:

$$n_R^{(0)}(x) = n_L^{(0)}(-x),\qquad(14)$$

which in turn implies $N_R = N_L$ for the total number of left- and right-movers, and also implies that the net flux is zero: $\phi = 0$. In contrast, an asymmetric distribution appears if the condition in Eq (14) is not satisfied. Instead of this rigorous definition of symmetry in terms of the detailed distribution, namely $n(x,t) = n(-x,t)$, we can just consider the symmetry in a weaker sense by demanding only the integral condition $N_R = N_L$. In such case the spatial distribution of ants may not look symmetric, but still we find the average position to be zero, $\bar{x}(t) = 0$, and the net flux to vanish as well, $\phi = 0$. In contrast, an asymmetric scenario will be one where $N_R \neq N_L$, and as a consequence $\bar{x} \neq 0$ and $\phi \neq 0$. Now, in the case that the velocity has a unique value, as in Eq (11), both definitions are equivalent [i.e. Eq (14) is fulfilled if and only if $N_L = N_R$]. Detailed description of the SSB formulation in (i) no-panic case (a symmetric escape) and (ii) panic case (an asymmetric escape) are shown in next Section.

## 5 Formulation of Panic escape in terms of SSB

### No panic case: a symmetric escape

Let us first consider an example of symmetric escape, i.e. the right- and left-moving densities satisfy $n_R^{(0)}(x) = n_L^{(0)}(-x)$ [see Fig 5]. In this example, these individual densities are not symmetric by themselves, but the sum is symmetric because Eq (14) is satisfied. Consequently $N_L = N_R$ for $t > 0$ and the symmetry is preserved.

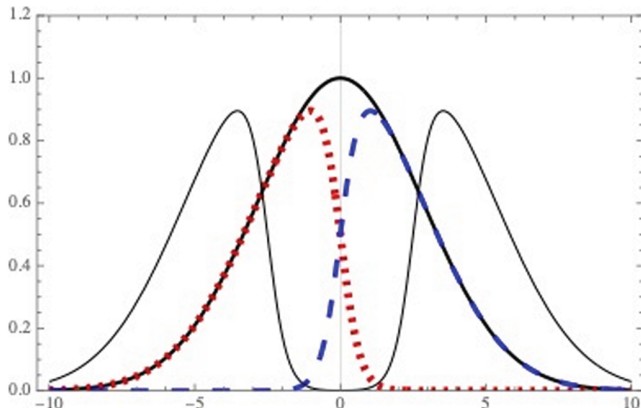

**Fig 5. Example of a symmetric distribution $n(x)$ at $t = 0$ (solid thick black) and at a later time (solid thin black).** The left-mover (red dotted) and right-mover (blue dashed) distributions $n_L^{(0)}(x)$ and $n_R^{(0)}(x)$ are also shown. The normalization and distance units are arbitrary.

In Fig 5 we use a skewed gaussian to represent the right- and left-mover densities:

$$n_R^{(0)}(x) = n_L^{(0)}(-x) \ = \ \frac{e^{-x^2/16}}{1 + e^{-3x}}, \tag{15}$$

neither of them symmetric around $x = 0$. However the total distribution at $t = 0$ is symmetric under $x \to -x$:

$$n(x) \equiv n_R^{(0)}(x) + n_L^{(0)}(x) \ = \ e^{-x^2/16}, \tag{16}$$

and at later times $t$:

$$n(x,t) \equiv n_R^{(0)}(x - vt) + n_L^{(0)}(x + vt) \tag{17}$$

it remains symmetric due to the condition $n_R^{(0)}(x) = n_L^{(0)}(-x)$, as it can be easily shown.

This is an example where the flux $\phi$ settles at $\phi_0 = 0$. Recall from Sect 3 that we have called $\phi_0$ the preferred flux, i.e. the flux that minimizes the cost function $V(\phi)$. For $\phi_0 = 0$ to be the value that minimizes the cost function, as shown in Sect 3, we should be in the regime where the density parameter is below the critical value, $\rho < \rho_c$.

In this regime, ants move to the right and left in the same amounts. According to experiments, such symmetric scenario would correspond to an escape where panic is not dominating the crowd.

## Panic case: An asymmetric escape

Now consider an asymmetric evolution. Within our approximation, this means necessarily $N_L \neq N_R$, which in turn means that the integrals of $n_R^{(0)}(x)$ and $n_L^{(0)}(x)$ must be different. Without loss of generality, let us assume $N_R > N_L$. Within the assumption that all the ants move at the same speed $v$, we can use a skewed gaussian to represent the right- and left-mover densities as

$$n_R^{(0)}(x) = \frac{e^{-x^2/16}}{1 + e^{-(x+\phi_0)}}, \quad n_L^{(0)}(x) = \frac{e^{-x^2/16}}{1 + e^{(x+\phi_0)}}, \tag{18}$$

where the skewedness is caused by the non-zero value of $\phi_0$. Consequently, the integral of $n_R^{(0)}(x)$ (blue dashed curve) is larger than the integral of $n_L^{(0)}(x)$ (red dotted curve), so that $N_R > N_L$. In Fig 6 we show an example of asymmetric distribution $n(x)$ with $\phi_0 = 3$.

With these right- and left-mover distributions, the total distribution at $t = 0$, $n(x) = n_R^{(0)}(x) + n_L^{(0)}(x)$ (solid thick black curve) is still symmetric: $n(x) = e^{-x^2/16}$. However, because of $N_R > N_L$, at later times the total distribution is no longer symmetric (solid thin black curve): there are more right-movers than left-movers. This asymmetry implies that now the flux, according to Eq (11), is non-vanishing. This flux should correspond to a $\phi_0 \neq 0$ that minimizes the cost function $V(\phi)$. As shown in Sect 3, this case should correspond to a density parameter higher than the critical value, $\rho > \rho_0$, and to a scenario of spontaneously broken symmetry: The transformation $x \to -x$ is still a symmetry of the laws of motion but it is no longer a symmetry of the state of minimum cost. We know it is the case of a hidden (or spontaneously broken) symmetry and not simply a case of no symmetry at all because, even when the state of minimum cost we found is not symmetric, there is a *degeneracy*: the symmetry transformation $x \to -x$ takes us from one state of minimum cost (flux $\phi = \phi_0$) to another state with equally minimum cost but opposite flux ($\phi = -\phi_0$).

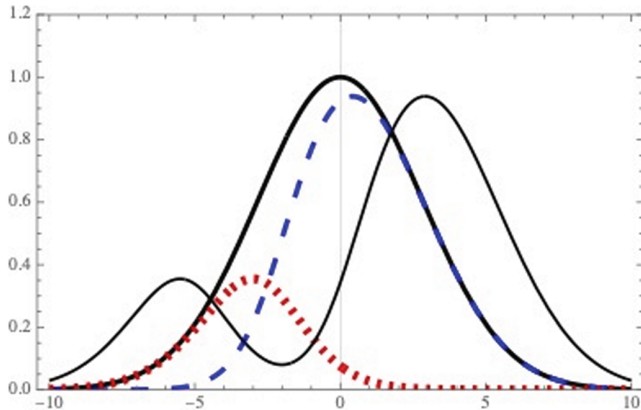

**Fig 6. Example of a symmetric distribution $n(x)$ at $t = 0$ (solid thick black) which is not symmetric at later times (solid thin black).** The integral of right movers $n_R^{(0)}$ (blue dashed) is larger than the integral of left movers $n_L^{(0)}$ (red dotted). Here we assumed $\phi_0 = 3$. The normalization and distance units are arbitrary.

## 6 Preventing asymmetrical panic-induced disasters

Panic situations can often have serious social consequences such as large-scale injuries or fatalities, sometimes requiring governmental institutions or other organizations to consider appropriate responses and preventive measures. Through measures such as appropriate architectural designs, exit planning, education, information dissemination, and crowd management, panic situations can be mitigated and prevented, reducing the risk of social disasters. Research and efforts to address panic play a vital role in protecting the safety and well-being of society as a whole.

Methods for preventing panic in general may include (1) *Exit and evacuation planning*: Large-scale events or crowded areas require clear plans and measures for exits and evacuation. Proper placement, the number of exits, and appropriate signage can help people safely evacuate during emergencies. (2) *Education and training*: Education and training on evacuation procedures and safety measures during emergencies are essential. Education and training can enhance understanding of panic situations and effective responses. (3) *Communication and information dissemination*: Rapid and effective information dissemination is crucial during emergencies. Establishing communication methods and providing real-time information to people is necessary in large-scale events or crowded areas. (4) *Crowd management and control*: Effective crowd management and control are essential in large-scale events or venues. Crowd management plays a significant role in preventing panic situations.

The important point here is that while the general view method mentioned above allows for qualitative understanding, it makes quantitative explanation difficult. It should be emphasized that, unlike this general view method, our approach allows for a quantitative approach. To thoroughly analyze the panic phenomena from a big-picture perspective, as per our original goal, we introduced two fundamental parameters: $\rho$ (internal density) and $f$ (external information). Through the relationship between $\rho$ and $f$, and the relationship between $\rho$ and its critical value $\rho_c$, we can quantitatively explain the panic phenomena. Of course, to apply this approach specifically and in detail to individual panic situations, the parameters $f$, $\rho$ and $\rho_c$ must be identified and specified for each case. However, to achieve our original goal of a complete analysis from a big-picture perspective, only a very small number of parameters are needed. For instance, in panic scenarios, specific combinations of two parameters,

$f$ and $\rho$, lead to SSB scenarios, i.e. scenarios of asymmetric escape where the possibility of panic-induced disasters are more likely. Going back to the SSB scenarios where $\rho > \rho_c$, from Eq (5) we see that the asymmetry is stronger (larger $\phi_0$) as $\rho$ gets larger (more influence of neighbors) and $f$ gets smaller (less external information).

Referring to the stable or minimal cost solution $\phi = 0$ (symmetric case) or $\phi = \pm\phi_0$ (asymmetric case) provided in Eq (5) and the graph depicting the cost function $V(\phi)$ shown in Fig 4, it becomes evident that asymmetric escape scenarios occur when $\rho > \rho_c$. In other words, there must be a critical density value, $\rho_c$, above which the panic escape becomes asymmetric and the influence of near neighbors dominate, disregarding or ignoring external information. The actual value of $\rho_c$ will depend on the system, and it could be even zero, meaning that the symmetry will be spontaneously broken for any $\rho$. These are the SSB scenarios where panic dominates. In contrast, for densities below $\rho_c$, the escape is symmetric, where we interpret that the external information is properly followed and the escape is safer.

As shown by the ant experiment, the escape situation may lead to disaster if the escape is highly asymmetric. Therefore, in order to prevent the disaster, ideally one should restrict the density of individuals $\rho$ to be below the critical value $\rho_c$, as long as that critical value is known. The critical value $\rho_c$ should depend on characteristics of the system–the shape of the place, the number of exits, the characteristics of the individuals including the interactions among them, and so on. Therefore, if the system can be designed beforehand (i.e. a building, a stadium, a neighborhood), the best way to guarantee safety in case of panic situations is to design the place so that a value of $\rho_c$ as large as possible is attained. Recall that the density combination $(\rho_c - \rho)$ is the coefficient of the quadratic term in the cost function $V(\phi)$ [see Eq (4)]. In our SSB formulation, we interpret that the symmetric case, i.e. $\phi = 0$, is the desirable state of the system in case of panic, because it corresponds to the minimum cost given by the quartic term (the term proportional to $f$), which represents the external information, i.e. the information or recommendation of the authorities or experts in case of panic situations. The quadratic term instead, represents the influence of near neighbors in the crowd. The denser the crowd is (larger $\rho$), the stronger the influence on each individual from the behavior of his/her neighbors. Now $\rho_c$ is the threshold value for $\rho$: if $\rho$ is lower, the influence of the neighbors is still pointing to a symmetric escape, i.e. a reaction similar to what the external rules imply. Instead, if $\rho$ is above this threshold, the influence of the neighbors is stronger, to the level that the individuals tend to disregard the external rules and spontaneous symmetry breaking (SSB) sets in. However, even in the SSB regime, the effect may not lead to disaster (i.e. $\phi_0$ small enough) if the external rules are strong enough (i.e. large $f$): as shown in Eq (5), $\phi_0 \to 0$, in the case of $(\rho - \rho_c)/f \to 0$.

In cases where the value of $\rho_c$ cannot be determined or when it is estimated to be zero, the prevention of disasters is achieved by increasing $f$ as well as keeping $\rho$ low. As shown above, even if there is spontaneous asymmetry in the escape, the size of the asymmetry (i.e the value of $\phi_0$) is reduced if $f$ is larger.

Finally we note that the SSB scenario could only be observed in systems which have a priori a symmetry, that could be broken spontaneously. As described before, in a panic situation where the geometry (*i.e.* boundary condition) is non-symmetric from the start, *e.g.* a room where there is only one door, or a symmetric room with two opposite doors but where a fire occurs on one of them, it is not possible to study a phenomenon of SSB due to panic [52], because there is no symmetry to be broken from the start. Any specific geometry and/or boundary condition are actually one of the ways to enforce external rules instead of imitation of nearest neighbor behavior. For example, in some spaces that gather large crowds of public, such as a stadium or a concert hall, some designers consider pillars or

obstacles to disrupt the crowd behavior of imitating neighbors and force the escapees to pay attention to the external rules. Our final goal is not to study the SSB, but to use the SSB to try to determine why a panic situation could cause it, and what kinds of parameters are determining the SSB scenario. Even in panic situations, where seemingly the SSB could occur, if either the values of $\phi_0$ and/or $(\rho - \rho_c)/f$ approach to zero, then the scenario remains symmetric [28,52].

Therefore, in order to avoid panic disasters, two features can be designed beforehand: (a) an effective set of external rules or information to be followed in case of panic, rules that must be clear and well understood by the individuals and (b) a design of the space that can ensure a large value for the density threshold $\rho_c$. Besides that, the population density $\rho$ should be kept below the threshold at any time if at all possible.

## 7 Conclusion

We have described panic escape scenarios based on an experiment with ants, where disaster is caused by an undesirable reaction of the crowd, which here is identified as an asymmetric stampede in an otherwise symmetric system. Without dealing with specific details of the individuals or the environment, we conjecture that the onset of dangerous panic reactions can be formulated in terms of the general mechanism of Spontaneous Symmetry Breakdown (SSB). This formulation is quite simple, allowing us to identify the main features of the system that can lead to disasters in panic situations, as the parameters of the formulation that lead to the onset of SSB. First is the order parameter, the parameter that exhibits SSB, which here is identified as the average flux during the escape. Then we have $f$, the coefficient of the quartic term in the cost function, which represents the external information (the external instructions or rules that should be followed in case of panic situations). Finally we have $\rho$, which represents the density of the crowd and which has a threshold value $\rho_c$ that separates the symmetric regime ($\rho < \rho_c$) from the SSB regime ($\rho > \rho_c$). Considering that the risk of disaster is higher the more asymmetrical the escape is (i.e. the larger the asymmetric flux), in order to avoid such scenario $\rho_c$ and $f$ should be as large as possible and $\rho$ should be kept below the threshold $\rho_c$. Specifically, $f$ increases with better information and instructions in case of panic situations, that should be easily assimilated and clearly followed by the crowd. In turn, $\rho_c$ depends on design features of the site, such as the shape of the place, the number of exits, and so on. It also could depend on characteristics of the individuals such as culture, habits, tendency to follow others, and so on. The parameters $\rho_c$ and $f$ should be considered beforehand, at the moment of design of the site, while a low enough value of $\rho$ is a condition that must be considered or enforced at all times. Since our study is only trying to establish the conjecture that the SSB should be a guiding principle in panic escape situations, we cannot provide yet numbers for the critical values of the onset of the SSB at each specific case. It is possible that there exist universal numbers that depend on general characteristics of the system such as dimensionality of the space (as it occurs in condensed matter physics), but that study is beyond this work.

In summary, we have introduced a novel approach based on the mechanism of *spontaneous symmetry breaking* to describe the onset of potentially disastrous outcomes in panic-driven escape situations. Its justification is empirical, based on experiments done with ants. Its advantage is that it contains few but important parameters that determine the onset of the critical scenarios, parameters that in principle can be controlled. Based on these parameters, we conclude strategies for preventing asymmetrical panic-induced escape scenarios that could lead to social disasters.

## Author contributions

**Conceptualization:** Choong Sun Kim, Claudio Dib, Sechul Oh.

**Formal analysis:** Choong Sun Kim, Claudio Dib, Sechul Oh.

**Investigation:** Choong Sun Kim, Claudio Dib, Sechul Oh.

**Methodology:** Choong Sun Kim, Claudio Dib, Sechul Oh.

**Supervision:** Choong Sun Kim, Claudio Dib, Sechul Oh.

**Validation:** Choong Sun Kim, Claudio Dib, Sechul Oh.

**Writing – original draft:** Choong Sun Kim, Claudio Dib, Sechul Oh.

**Writing – review & editing:** Choong Sun Kim, Claudio Dib, Sechul Oh.

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
