## [Decision Letter · Decision Letter 0]

20 Feb 2025

PONE-D-24-48603Spontaneous Symmetry Breaking and Panic EscapePLOS ONE

Dear Dr. OH,

Thank you for submitting your manuscript to PLOS ONE. After careful consideration, we feel that it has merit but does not fully meet PLOS ONE’s publication criteria as it currently stands. Therefore, we invite you to submit a revised version of the manuscript that addresses the points raised during the review process.

We look forward to receiving your revised manuscript.

Kind regards,

Jiankun Gong

Academic Editor

PLOS ONE

Journal Requirements:

NRF of Korea (NRF-2022R1A5A1030700 and NRF-2022R1I1A1A01055643). Chile grants Fondecyt 1210131 and ANID PIA/APOYO AFB230003.

The work of CSK is supported by NRF of Korea (NRF-2022R1A5A1030700 and NRF-2022R1I1A1A01055643). We acknowledge support from Chile grants Fondecyt 1210131 and ANID PIA/APOYO AFB230003.

NRF of Korea (NRF-2022R1A5A1030700 and NRF-2022R1I1A1A01055643). Chile grants Fondecyt 1210131 and ANID PIA/APOYO AFB230003.

Reviewers' comments:

Reviewer's Responses to Questions

**Comments to the Author**

1. Is the manuscript technically sound, and do the data support the conclusions?

Reviewer #1: Yes

Reviewer #2: Yes

Reviewer #3: Yes

2. Has the statistical analysis been performed appropriately and rigorously? 

Reviewer #1: Yes

Reviewer #2: Yes

Reviewer #3: N/A

3. Have the authors made all data underlying the findings in their manuscript fully available?

Reviewer #1: Yes

Reviewer #2: Yes

Reviewer #3: Yes

4. Is the manuscript presented in an intelligible fashion and written in standard English?

Reviewer #1: Yes

Reviewer #2: Yes

Reviewer #3: Yes

5. Review Comments to the Author

Reviewer #1: The article presents a simple formulation to model the movement of subjects under observation and propose parameters to evaluate the symmetry of of this enclosed system. The main objective of the work is to reduce the variables in modelling the panic-induced escape using the many-particle system. Some comments regarding the submitted work are:

1. The model considers one-dimensional movement of the subjects, however would the application of the model remains intact if the multi-dimensional movement is considered? and what if there are odd number of escape routes?

2. The movement of subjects is measured using group velocity, which in return would depend upon external factors such a heat, smoke, flood, age of the subjects, and other similar variables in the practical scenario. Authors may please comment if the group velocity as a single dependent variable suffice all such independent variables.

3. How can one incorporate the boundary conditions in the given model? assuming the space considered in the exiting model is homogenous and unidimensional.

4. It would be beneficial to the readers if authors can provide a numeric figure for the critical population density or a relative parameter depending upon population size.

Overall the paper is well-written and can be accepted with the suggested revisions.

Reviewer #2: This paper presents a novel approach to describe escape type panic. However, the symmetry description of the escape panic scene and the description of the symmetry breaking conditions are not clear enough, so it is suggested to improve them.

Reviewer #3: This study aims to analyze and formulate panic phenomena in disaster evacuation behavior using the concept of Spontaneous Symmetry Breaking (SSB) from physics. The motivation for this approach is that existing models for evacuation and panic behavior, such as the Social Force Model, require many parameter estimations and are developed individually for specific scenarios like fires, earthquakes, and floods. Therefore, a general model that is simple (with fewer parameters) and can cover multiple phenomena is needed. The study suggests that this can be achieved by viewing the phenomenon as SSB occurring in the physical system of a crowd of evacuees, rather than focusing on individual behavior.

I found this paper very interesting, but I didn’t realize its significance when I first read it. Due to the questions and lack of explanations mentioned below, I am concerned that researchers in the field of evacuation may not appreciate its value.

1. Panic and SSB

The authors of this paper attempt to generalize panic behavior using Symmetry Breaking (SSB), but their argument lacks persuasiveness. They mention the Social Force Model and the ant model by Altshuler et al. [12] as models of panic behavior, primarily using the ant model to explain SSB phenomena in evacuations. However, the discussion linking the ant model to human panic behavior during evacuations is insufficient.

In the “Possible sources of panic” section, many studies on panic behavior are cited, but can all of these be generalized by SSB? Has SSB been observed in all these studies?

For example, in [28], Haghani states that symmetry breaking and herding phenomena were confirmed as positive in only 4 out of 13 empirical studies on human evacuation behavior.

Furthermore, in paper [A], the occurrence of symmetry breaking phenomena in human evacuations is considered negative, suggesting that we should not easily link the symmetry breaking phenomena observed in ants to human evacuation behavior.

Based on these studies, the “Fundamental solution to these issues” section requires a more careful discussion to connect human panic behavior with the ant model.

[A] Milad Haghani and Majid Sarvi, 'Herding' in direction choice-making during collective escape of crowds: How likely is it and what moderates it ?, Safety Science 115 (2019), 362-375

2. Model and Parameter

Many researchers in evacuation studies view symmetry breaking as a phenomenon that emerges from evacuation behavior. In other words, whether in panic or not, the behavior causes symmetry breaking as a result, not the other way around. Naturally, evacuees do not intentionally cause symmetry breaking; it occurs as a phenomenon even though there is nothing in their behavior that causes asymmetry.

However, in this paper, it seems to be the opposite. Symmetry breaking is always assumed first, and the phenomenon is then retrospectively applied to evacuation. The parameters and models also appear to be chosen arbitrarily (or is that the intention?). For example, equation (4) is introduced to describe SSB, but it is not given any meaning in the context of evacuation phenomena. Despite this, the parameters used, such as Rho for population density and f for external information, are given meanings related to evacuation. These choices seem arbitrary, and there is no explanation for why the opposite choices would not work.

Such discussions might be common in physics, but they are unfamiliar and difficult to understand in the context of evacuation research.

3. order parameter Phi

In the “Panic escape in terms of SSB” section, the important parameter Phi in equation (4) is explained. Phi is defined as the net flux of ant behavior in equation (9), which is the total number of ants multiplied by their velocity. Since all ants are simplified to move at the same speed, it can essentially be considered a distribution (denoted as n(x) in this section). Equation (10) then expresses n(x) as nR(x) + nL(x), with detailed analysis provided in Appendix A.

Here, the explanations are divided into A.1 No panic case and A.2 Panic case. Although both cases result in the same total (A2), the subtle differences in the elements described by equations (A1) and (A4) lead to symmetry in the former and asymmetry in the latter. The slight difference in the denominator of exp(x+3) in (A1) and (A4) causes this effect.

However, this part is challenging to connect with evacuation phenomena. It would be helpful to discuss more concretely how specific evacuation phenomena relate to (A1) and (A4) and how parameters like Rho and f are connected.

The paper presents an abstract and theoretical discussion of evacuation phenomena as a physical model of SSB, but it would benefit from explanations that link these equations and parameter differences to specific evacuation examples. The parameter discussion in section 5 is too general and obvious to be connected to specific phenomena.

The discussion in Appendix A is interesting and might be better included in the main body of the paper.

In conclusion, while the paper is very intriguing and has the potential to bring new perspectives to evacuation research, it may be difficult for the evacuation research community to accept it in its current form. Providing more concrete examples would help demonstrate its usefulness.

6. PLOS authors have the option to publish the peer review history of their article (what does this mean?). If published, this will include your full peer review and any attached files.

Reviewer #1: No

Reviewer #2: No

Reviewer #3: **Yes: **Akira Tsurushima

---

## [Author Response · Author response to Decision Letter 1]

27 Mar 2025

PONE-D-24-48603

Response to reviewers for the manuscript entitled: “Spontaneous Symmetry Breaking and Panic Escape”, by C.S. Kim, C. Dib and S. Oh:

We thank the reviewers for their valuable comments and observations, which we have considered in this new version of our manuscript.

In what follows, we address each point raised by the academic editor and reviewers. First, we provide a summary of the changes made to the original version in the revised manuscript, and then we answer each point raised by the reviewers.

The summary of changes in the manuscript is listed here (in the new manuscript, added or corrected text appears in blue):

[Revision #1] Page 7: just below Eq. (5), we added a sentence starting with “and phi_0 = 0 when ....”

[Revision #2] Page 7: just above the paragraph starting with “In summary, in the SSB ....”, we added a new paragraph starting with “Notice that we are considering an example ....”

[Revision #3] Page 7: we rephrased the beginning of the paragraph “In summary, in the SSB case the physical laws in this system,” by the text “In summary, in a SSB scenario, the physical laws of the system,”…

[Revision #4] Page 8: At the end of the paragraph starting with “Before we further describe the situation...”, we added five consecutive sentences starting with the sentence “Notice that many evacuation studies....” and ending with “…otherwise symmetric environments.”

[Revision #5] Page 9: we added a footnote starting with “A continuous distribution of speeds ....”

[Revision #6] Pages 10 and 11: we moved the discussion of Appendix A of the original version into the main body of the article.

[Revision #7] Pages 10 and 11: in subsection “Panic case”, at the end of the first paragraph which starts with: ”Now consider an asymmetric evolution…”, we added three consecutive sentences “Within the assumption that ... …with phi_0 = 3.”

[Revision #8] Page 11: in the caption of Fig. 6, we added two consecutive sentences “Here we assumed .... …are arbitrary.”

[Revision #9] Page 13: following the paragraph starting with “In cases where the value of ....”, we added a new paragraph starting with “Finally we note that ....”

[Revision #10] Pages 13 and 14: At the end of the first paragraph of the section Conclusion, we added two consecutive sentences “Since our study is .... …beyond this work.”

[Revision #11] Page 17: we added the new reference [52].

In what follows, we respond to each point raised by the reviewers. (Our answers are highlighted in blue, in the separate rebuttal letter file labeled 'Response to Reviewers'.)

Reviewer #1:

The article presents a simple formulation to model the movement of subjects under observation and propose parameters to evaluate the symmetry of this enclosed system. The main objective of the work is to reduce the variables in modelling the panic-induced escape using the many-particle system. Some comments regarding the submitted work are:1. The model considers one-dimensional movement of the subjects, however would the application of the model remains intact if the multi-dimensional movement is considered? and what if there are odd number of escape routes?

ANSWER: We consider an example of 1-dimensional SSB, where the options are necessarily discrete (escape left or right). However the generalization to more dimensions or a continuous set of escape options (i.e. any direction of 360º) in a model of SSB is straightforward. While it would be very contrived to conceive a physical layout where escape can be in any direction and SSB would be a problem (e.g. escape from a dangerous spot in an open field) the dynamics of SSB is still the same: does the crowd obey external instructions or is it dominated by the behaviour of nearest neighbors?

( In response to this point, we have revised the manuscript accordingly as shown in [Revision #2] of the summary of our changes. )

2. The movement of subjects is measured using group velocity, which in return would depend upon external factors such as heat, smoke, flood, age of the subjects, and other similar variables in the practical scenario. Authors may please comment if the group velocity as a single dependent variable suffice all such independent variables.

ANSWER: A continuous distribution of speed would characterize better some situations than others, but the main feature that SSB is addressing remains the same: does the crowd follow external pre-established rules or is it dominated by nearest neighbor behavior? The external factors that cause the panic escape we are assuming are not asymmetric (i.e. there is no fire on one specific side, because then obviously the individuals will run in the opposite direction). We are assuming the external factors that cause the panic are “symmetric”. The escape velocity, instead, is the response of the crowd to the panic situation, where the danger is not coming from a specific direction that will break the symmetry externally. Otherwise, we will be in a scenario that is not symmetric from the start, something which is known as “explicit”, not “spontaneous” symmetry breaking.

( In response to this point, we have revised the manuscript accordingly as shown in [Revision #5] of the summary of our changes. )

3. How can one incorporate the boundary conditions in the given model? assuming the space considered in the exiting model is homogenous and unidimensional.

ANSWER: Boundary conditions are actually one of the ways to enforce external rules instead of imitation of nearest neighbor behavior. For example, in some spaces that gather large crowds of public, such as a stadium or a concert hall, some designers consider pillars or obstacles to disrupt the crowd behavior of imitating neighbors and force the escapees to pay attention to the external rules.

( In response to this point, we have revised the manuscript accordingly as shown in [Revision #9] of the summary of our changes. )

4. It would be beneficial to the readers if authors can provide a numeric figure for the critical population density or a relative parameter depending upon population size.

ANSWER: Since our study is only trying to establish the conjecture that SSB should be a guiding principle in panic escape situations, we cannot provide yet numbers for the critical values of the onset of SSB. It is possible that there exist universal numbers that depend on general characteristics of the system such as dimensionality of the space (as it occurs in condensed matter physics), but that study is beyond this work.

( In response to this point, we have revised the manuscript accordingly as shown in [Revision #10] of the summary of our changes. )

Reviewer #2:

This paper presents a novel approach to describe escape type panic. However, the symmetry description of the escape panic scene and the description of the symmetry breaking conditions are not clear enough, so it is suggested to improve them.

ANSWER: To address this point, we have added several relevant discussions accordingly in the revised manuscript, including [Revision #2], [Revision #3], [Revision #4], [Revision #5], [Revision #6], [Revision #7], [Revision #8], [Revision #9] and [Revision #10] of the summary of our changes.

Reviewer #3:

This study aims to analyze and formulate panic phenomena in disaster evacuation behavior using the concept of Spontaneous Symmetry Breaking (SSB) from physics. The motivation for this approach is that existing models for evacuation and panic behavior, such as the Social Force Model, require many parameter estimations and are developed individually for specific scenarios like fires, earthquakes, and floods. Therefore, a general model that is simple (with fewer parameters) and can cover multiple phenomena is needed. The study suggests that this can be achieved by viewing the phenomenon as SSB occurring in the physical system of a crowd of evacuees, rather than focusing on individual behavior.I found this paper very interesting, but I didn’t realize its significance when I first read it. Due to the questions and lack of explanations mentioned below, I am concerned that researchers in the field of evacuation may not appreciate its value.

1. Panic and SSB

The authors of this paper attempt to generalize panic behavior using Symmetry Breaking (SSB), but their argument lacks persuasiveness. They mention the Social Force Model and the ant model by Altshuler et al. [12] as models of panic behavior, primarily using the ant model to explain SSB phenomena in evacuations. However, the discussion linking the ant model to human panic behavior during evacuations is insufficient.

ANSWER: Actually, the ant experiment already is establishing the link of an escaping crowd of ants with an escaping crowd of any kind of individuals, so in this paper we are following the same analogy.

In the “Possible sources of panic” section, many studies on panic behavior are cited, but can all of these be generalized by SSB? Has SSB been observed in all these studies?

For example, in [28], Haghani states that symmetry breaking and herding phenomena were confirmed as positive in only 4 out of 13 empirical studies on human evacuation behavior.

Furthermore, in paper [A], the occurrence of symmetry breaking phenomena in human evacuations is considered negative, suggesting that we should not easily link the symmetry breaking phenomena observed in ants to human evacuation behavior.

Based on these studies, the “Fundamental solution to these issues” section requires a more careful discussion to connect human panic behavior with the ant model.

[A] Milad Haghani and Majid Sarvi, 'Herding' in direction choice-making during collective escape of crowds: How likely is it and what moderates it ?, Safety Science 115 (2019), 362-375

ANSWER: The SSB scenario could only be observed in systems which have a priori a symmetry, that could be broken spontaneously. In a panic situation where the geometry is non-symmetric from the start, e.g. a room where there is only one door, or a symmetric room with two opposite doors but where a fire occurs on one of them, it is not possible to study a phenomenon of SSB due to panic, because there is no symmetry to be broken from the start. Our final goal is not to study SSB, but to use SSB to try to determine why a panic situation could cause it, and what kinds of parameters are determining the SSB scenario.

( In response to this point, we have revised the manuscript accordingly as shown in [Revision #9] of the summary of our changes. We have also added the paper [A] as the new reference [52] in the revised version of the manuscript as shown in [Revision #11].)

2. Model and Parameter

Many researchers in evacuation studies view symmetry breaking as a phenomenon that emerges from evacuation behavior. In other words, whether in panic or not, the behavior causes symmetry breaking as a result, not the other way around. Naturally, evacuees do not intentionally cause symmetry breaking; it occurs as a phenomenon even though there is nothing in their behavior that causes asymmetry.

However, in this paper, it seems to be the opposite. Symmetry breaking is always assumed first, and the phenomenon is then retrospectively applied to evacuation. The parameters and models also appear to be chosen arbitrarily (or is that the intention?). For example, equation (4) is introduced to describe SSB, but it is not given any meaning in the context of evacuation phenomena. Despite this, the parameters used, such as Rho for population density and f for external information, are given meanings related to evacuation. These choices seem arbitrary, and there is no explanation for why the opposite choices would not work.

Such discussions might be common in physics, but they are unfamiliar and difficult to understand in the context of evacuation research.

ANSWER: That is precisely the point: crowds have a tendency to follow the behavior of neighbors, even if there is no panic. This behavior will cause asymmetry even when the geometry of the place is symmetric. It is precisely the conjecture of our study: up to what point the crowd would follow neighbor behavior instead of external rules? In panic situations this behavoir could lead to disaster, but in safe situations it has no bad implications, certainly.

( In response to this point, we have revised the manuscript accordingly as shown in [Revision #4] of the summary of our changes. )

3. order parameter Phi

In the “Panic escape in terms of SSB” section, the important parameter Phi in equation (4) is explained. Phi is defined as the net flux of ant behavior in equation (9), which is the total number of ants multiplied by their velocity. Since all ants are simplified to move at the same speed, it can essentially be considered a distribution (denoted as n(x) in this section). Equation (10) then expresses n(x) as nR(x) + nL(x), with detailed analysis provided in Appendix A.

Here, the explanations are divided into A.1 No panic case and A.2 Panic case. Although both cases result in the same total (A2), the subtle differences in the elements described by equations (A1) and (A4) lead to symmetry in the former and asymmetry in the latter. The slight difference in the denominator of exp(x+3) in (A1) and (A4) causes this effect.

However, this part is challenging to connect with evacuation phenomena. It would be helpful to discuss more concretely how specific evacuation phenomena relate to (A1) and (A4) and how parameters like Rho and f are connected.

ANSWER: We did not try to establish dynamical equations that lead to the SSB situation. The dynamical laws should be able to determine the critical value rho_c. We did not do that. We only stated an example of how an original symmetric situation separates into asymmetrical motion.

For better clarity, we have adjusted (A4) by introducing “phi_0 = Sqrt{(rho-rho_c)/f}” into the equation, whose non-zero value causes the skewedness.

( In response to this point, we have revised the manuscript accordingly as shown in [Revision #7] and [Revision #8] of the summary of our changes. )

The paper presents an abstract and theoretical discussion of evacuation phenomena as a physical model of SSB, but it would benefit from explanations that link these equations and parameter differences to specific evacuation examples. The parameter discussion in section 5 is too general and obvious to be connected to specific phenomena.

ANSWER: We are assuming an initial condition where the population is spread in space as a Gaussian (normal) statistical distribution. The initial condition could be any other configuration, but we chose this one as an example of starting point. From then on, the dynamics should set the net flux phi to be zero or non-zero. We try to characterize how the initial distribution would evolve if the flux is symmetric or if it is not symmetric.

( In response to this point, we have revised the manuscript accordingly as shown in [Revision #7] and [Revision #8] of the summary of our changes. )

The discussion in Appendix A is interesting and might be better included in the main body of the paper.

ANSWER: We agree. It was a point we were not sure. So we now put this discussion of Appendix A into the main body.

( In response to this point, we have revised the manuscript accordingly as shown in [Revision #6] of the summary of our changes. )

In conclusion, while the paper is very intriguing and has the potential to bring new perspectives to evacuation research, it may be difficult for the evacuation research community to accept it in its current form. Providing more concrete examples would help demonstrate its usefulness.

Thank you very much.

---

## [Editor Report · Decision Letter 1]

30 Mar 2025

Spontaneous Symmetry Breaking and Panic Escape

PONE-D-24-48603R1

Dear Dr. OH,

We’re pleased to inform you that your manuscript has been judged scientifically suitable for publication and will be formally accepted for publication once it meets all outstanding technical requirements.

Kind regards,

Jiankun Gong

Academic Editor

PLOS ONE
---

## [Editor Report · Acceptance letter]

PONE-D-24-48603R1

PLOS ONE

Dear Dr. OH,

I'm pleased to inform you that your manuscript has been deemed suitable for publication in PLOS ONE. Congratulations! Your manuscript is now being handed over to our production team.

Kind regards,

on behalf of

Dr. Jiankun Gong

Academic Editor

PLOS ONE